# Surgery for Xanthogranulomatous Pyelonephritis: A Comparison of Midline Transperitoneal and Flank Retroperitoneal Laparotomy Approaches to Nephrectomy

**DOI:** 10.3390/jcm11154476

**Published:** 2022-07-31

**Authors:** Shu-Han Tsao, Chien-Ho Wang, Horng-Heng Juang, Yu-Hsiang Lin, Pei-Shan Yang, Phei-Lang Chang, Chien-Lun Chen, Chen-Pang Hou

**Affiliations:** 1Department of Urology, Chang Gung Memorial Hospital at Linkou, Taoyuan 333, Taiwan; terry10657@hotmail.com (S.-H.T.); hhj143@mail.cgu.edu.tw (H.-H.J.); linyh@cgmh.org.tw (Y.-H.L.); dr.yang.uro@gmail.com (P.-S.Y.); henryc@cgmh.org.tw (P.-L.C.); clc2679@cgmh.org.tw (C.-L.C.); 2Deartment of Emergency Medicine, Keelung Chang Gung Memorial Hospital, Keelung 204, Taiwan; a151802@outlook.com; 3School of Medicine, Chang Gung University, Taoyuan 333, Taiwan; 4Department of Anatomy, School of Medicine, Chang Gung University, Kwei-Shan, Taoyuan 333, Taiwan; 5Graduate Institute of Clinical Medical Sciences, College of Medicine, Chang Gung University, Taoyuan 333, Taiwan; 6Department of Healthcare Management, Yuanpei University of Medical Technology, Hsinchu 300, Taiwan

**Keywords:** xanthogranulomatous pyelonephritis, XGP, nephrectomy, transperitoneal, retroperitoneal

## Abstract

Xanthogranulomatous pyelonephritis (XGP) is a rare inflammatory disease often associated with high morbidity and mortality. Whether the midline transperitoneal or the flank retroperitoneal approach is superior remains unknown. We searched through pathology databases and reviewed 86 patients with an XGP diagnosis from 2000 to 2021 at our institution. After the patients who did not meet the inclusion criteria were excluded, 35 patients who had undergone nephrectomy through the midline transperitoneal or the flank retroperitoneal laparotomy approach were recruited. Nine (25.71%) of the thirty-five patients underwent nephrectomy through a midline approach, whereas twenty-six (74.29%) received a flank approach. Patients in the midline approach group had a longer surgical time (*p* = 0.03) than those in the flank approach group. In addition, patients in the flank approach group took less time after surgery to resume oral intake than those in the midline approach group (*p* = 0.01). No significant differences in the rates of intraoperative and postoperative complications such as peritonitis or intraabdominal infection were observed between the groups. For the patients with XGP who are good candidates for surgery, nephrectomy is a relatively safe surgical treatment method. Both surgical methods produced favorable surgical outcomes, and the patients who received these methods had similar complication rates.

## 1. Introduction

Xanthogranulomatous pyelonephritis (XGP) is an uncommon but serious condition in which the normal kidney parenchymal can be replaced by lipid-laden macrophages or histiocytes [1]. It accounts for only 0.6 percent of histologically documented cases of pyelonephritis, according to the previously published data [2]. Urinary obstruction, infection, kidney stones, diabetes, and immunocompromised states are all common causes of XGP [1]. Patients with XGP frequently have flank pain, high fever, anorexia, and weight loss. Chronic febrile sickness, urosepsis, and even nephrogenic hepatic dysfunction may also manifest in the affected patients [2]. It can spread to the surrounding tissue if left untreated and may require surgery in extreme cases to prevent further deterioration of renal function [2,3,4,5]. Nephrectomy is highly challenging as XGP is a very severe infectious condition. According to the report by Malek et al., only 80% of the patients were able to undergo the full procedure; the remaining patients either underwent only biopsy or only excision of the diseased renal segment [2].

Although nephrectomy is the definitive treatment for a severe type of XGP, it has a 21.9% complication and a 4.9% mortality rate [5]. Such surgery could potentially result in significant complications, including septic shock, intestinal damage, and damage to the inferior vena cava; in a previous report, two patients died in the immediate early postoperative period of septic shock [5]. Studies have indicated that patients can receive minimally invasive surgery only if their condition is relatively uncomplicated [5,6,7,8,9,10,11]. The benefits of minimally invasive surgery apply only to suitable candidates. However, some patients, particularly those with rugged or bulky kidneys, are not suitable candidates; conventional laparotomy is the best option for them [12]. Laparotomy is less hazardous and produces better outcomes in some complicated XGP cases [13,14,15]. However, the evidence for deciding between the approaches for open nephrectomy is lacking. A thorough review of surgical techniques is warranted for a condition with a significant risk of complications and fatality. As a result, this study examined the surgical outcomes of the midline transperitoneal and flanked retroperitoneal approaches in patients undergoing nephrectomy for XGP.

## 2. Materials and Methods

### 2.1. Study Design

We performed a retrospective review by examining the pathological results of kidney biopsy specimens extracted between 2000 and 2021 at Linkou Chang Gung Memorial Hospital in Taiwan. All the patients with histologic evidence of XGP were included. This retrospective review study was approved by the Chang Gung Medical Foundation Institutional Review Board (IRB: 202200639B0). Because of the retrospective nature of this study, the above-mentioned committee approved the request to waive informed consent.

### 2.2. Patients

All the patients who underwent nephrectomy with pathologic evidence of XGP were included. The patients who had missing medical records, had a diagnosis of concomitant cancer, or did not undergo a unilateral open nephrectomy were excluded. The screening process for patients in our study is presented in Figure 1. The patients were divided into two groups on the basis of surgical approach: midline transperitoneal (group I) or flank retroperitoneal (group II).

### 2.3. Surgical Approaches

Surgical methods were selected after discussion between the surgeons and the patients. The patients’ positions and the directions through which the surgeons accessed the abdominal cavity varied between the groups. For the midline approach, the whole peritoneum and retroperitoneum were accessed by making a vertical midline incision along the xiphoid process to just below the umbilicus. Then, after the colon was dissected, the retroperitoneal space entered, and the pedicle regulated, nephrectomy was performed. As for the flank approach, the retroperitoneal cavity was accessed via a flank incision over the 11th or 12th rib regardless of whether partial ribs had been removed. Nephrectomy was performed without entering the abdominal cavity by controlling the pedicle.

### 2.4. Data Collection

We collected demographic features (Table 1), namely sex, age, body mass index (BMI), comorbidity, the American Society of Anesthesiologists Classification (ASA Classification) [16], preoperative laboratory data, initial clinical manifestations, preoperative diagnosis, preoperative drainage placement, and the Malek and Elder classification [4].

### 2.5. Outcomes

The primary outcome was perioperative status, namely the operative duration, kidney specimen volume, and the Clavien–Dindo classification of complications [17]. The secondary outcome was postoperative recovery status, namely postoperative drainage placement time, analgesic use, antibiotic duration, length of hospitalization after surgery, and time to resume oral intake.

### 2.6. Statistical Analysis

The continuous variables for baseline demographic features were expressed as means ± standard deviations (SD), whereas the categorical data were presented as percentages. To examine the differences in baseline demographic features between the groups, the continuous and categorical data were analyzed using independent t tests and chi-squared tests, respectively. A two-tailed *p* value of less than 0.05 was considered statistically significant. All analyses were performed using IBM SPSS Statistics version 24.0 (SPSS Inc., Chicago, IL, USA).

## 3. Results

We searched through a pathology database and identified 86 patients that tested positive for XGP between 2000 and 2021. Twelve patients were not included in the study because they had received a diagnosis before the electronic medical record system was developed, and their medical data were partly incomplete. A total of 64 individuals with an XGP diagnosis received surgery, with another 10 receiving a kidney biopsy. Eighteen of the patients received a diagnosis of cancer and were thus excluded. To compare the two surgical approaches, only the individuals who had received a unilateral open nephrectomy were included. We also excluded four patients who had received minimally invasive surgery, three who had received bilateral surgery, one who had received unroofing of the renal cyst, one who had angiomyolipoma, one who had a partial nephrectomy, and one who had received an adrenalectomy. The study included 35 patients who had undergone open nephrectomy (Figure 1).

A total of 77.1% of the patients had abdominal discomfort, the most common symptom among our patients. Additionally, 65.7% of the patients had a urinary tract infection, 42.9% had hematuria, and 34.3% had a fever. Stones were observed in 68.6%, hydronephrosis in 65.7%, and kidney malignancies in 22.9% on the basis of preoperative computed tomography. A midline approach to open nephrectomy was used in 9 of 35 patients (25.71%), whereas 26 (74.29%) received a flank approach. Table 1 summarizes the patients’ demographic and clinical characteristics. Apart from the ASA classification and the Malek and Elder categorization, no significant differences were observed between the groups. 

In both groups, no deaths occurred within 30 days of the surgical operation and recovery (Table 2). The patients undergoing nephrectomy using the midline approach had longer surgical times (*p* = 0.03) than did those undergoing nephrectomy using the flank approach (*p* = 0.01). In terms of the time to resume oral intake after surgery, the patients in the flank approach group required less time (*p* = 0.01) than did those in the midline approach group (*p* = 0.01). No significant differences were observed in the other postoperative outcome parameters, namely estimated blood loss, blood transfusion rate, kidney specimen volume, postoperative complication rate, duration of drainage placement, analgesic and antibiotics use, length of hospitalization, postoperative ICU care, and return to ER within 3 months.

No significant differences in major complication rates of the patients during and after surgery were observed (*p* = 0.45 and 0.38, respectively). In the midline group, one patient had spleen damage, and a splenectomy was performed simultaneously during the surgery. Another patient developed a pleural effusion after surgery and required chest echo tapping. In the flank group, pancreatitis was identified in one patient during surgery, and primary repair was performed by a general surgeon. Severe postoperative complications were identified in three patients in the same group; two had colon injury, one had an intra-abdominal abscess, and one had fistula. A retroperitoneal abscess requiring operation was observed in one patient after surgery. 

Table 3 presents the results of the urine and perinephric abscess cultures. Escherichia coli were the most common organism (51.4%), followed by Proteus mirabilis (31.4%). Multiple organisms were observed in the urine and perinephric abscess cultures of 40% of the patients. One patient in the midline group and three patients in the flank group had negative cultures.

## 4. Discussion

XGP usually appears as a kidney enlargement that mimics a neoplasm and is therefore known as a renal pseudotumor. The disorder is characterized by the disintegration of renal or perirenal tissue and its replacement by granulomatous tissue containing lipid-laden macrophages. The inflammatory process could extend to surrounding tissue if left uncontrolled [1]. For the kidney affected by XGP, nephrectomy might be the best treatment option, although medical treatment can be helpful and effective. However, in some cases, conventional laparotomy can outperform the laparoscopic approach [12]. However, no general consensus has been reached regarding whether laparotomy is the best option. We compared two different approaches to laparotomy; no differences in complication rates or length of hospitalization were observed. The flank approach led to shorter operation times and less time to resume oral intake than did the midline approach. 

Laparoscopic nephrectomy is in cancer surgery, with most results being similar to those of an open approach but some being better in terms of aesthetics and blood loss reduction [18,19]. However, the laparoscopic management of inflammatory kidney disease remains challenging because XGP is the trickiest type of pyelonephritis to handle. Laparoscopic surgery for XGP patients can frequently be switched to a laparotomy during surgery. According to Angeri et al., 40% of patients with XGP were switched to open surgery during laparoscopic surgery [12]. Several studies have also indicated that 7% to 50% of patients are switched to open surgery during laparoscopic surgery for various reasons, including the higher risk of complications [5,6,7,8,9]. Therefore, XGP nephrectomy is more complex and technically demanding than noninfectious nephrectomy. For a well-trained and experienced surgeon, the laparoscopic approach for XGP treatment is a viable surgical option [20]. However, the open approach is more suitable for inexperienced laparoscopists [9,12,21]. No study has compared the midline and flank approaches in the surgical treatment of XGP despite the fact that laparoscopic nephrectomy for XGP has a higher chance of being switched to open surgery. Further research on open surgery for XGP should be conducted to provide valuable clinical information.

The main difference between the surgical approaches is whether the abdominal cavity is penetrated. According to Taue et al., in terms of tumor size and location the transperitoneal approach should be used for laparoscopic radical nephrectomy to prevent vascular injury. In addition, the average weight of the resected specimen in the transperitoneal group was heavier than that in the retroperitoneal group [15]. In our study, the average resected specimen of the midline group weighed 531.9 g, which was heavier than the 338.7 g of the flank group. Therefore, the transperitoneal approach may be better suited for larger kidney specimens, although the difference was nonsignificant (*p* = 0.39). The midline transperitoneal technique provides more working space and is easier to interpret anatomically. A prospective randomized comparative study of laparoscopic radical nephrectomy discovered that retroperitoneoscopy may result in early renal hilar control and shorter overall operative times [13]. In our research, the flank retroperitoneal approach required significantly shorter operative times than did the midline transperitoneal approach. The advantages of the flank retroperitoneal approach are that the renal artery can be approached and that it has a lower risk of complications in the abdomen’s surrounding organs. Most studies, however, have indicated that the surgical method has little effect on the frequency of complications and postoperative outcomes [13,14,15]. 

In our study, the midline transperitoneal approach required longer operative times, but patients who received the flank retroperitoneal approach exhibited earlier oral intake. Although the length of hospitalization following surgery was shorter in the flank group, the differences did not reach statistical significance. Additionally, in terms of intraoperative and postoperative complication rates, no significant difference was observed between the two groups. The surgical complexity of the midline approach is thus higher than that of the flank approach, which could explain why the midline approach requires more time. Except for the longer operations and the time to resume post-surgery feeding in the midline approach, no complications were observed in the midline approach group. No significant differences in intraoperative or postoperative complications were observed between the two surgical methods. Considering that the surgeons’ expertise and knowledge of surgical methods may vary, the midline approach may be a safe surgical treatment option, especially for patients at higher risk, because it produces similar results to those of the flank approach. The flank method can also be considered because of the rapid postoperative recovery. Patients may therefore be able to begin intake earlier and be discharged earlier from the hospital.

Surgeons often worry whether the excision of infected kidneys through the transabdominal approach will result in peritonitis or the accumulation of intra-abdominal abscesses. Studies have indicated that treating pre-existing retroperitoneal abscesses with abdominal surgery increases the risk of complications such as peritonitis and septic shock [22]. Our investigation of the transperitoneal technique revealed no postoperative peritonitis or abscess formation. Adel et al. also observed that only 1 of 20 patients with transperitoneal laparoscopic pyelolithotomy exhibited peritonitis due to stones left in the abdominal cavity [23]. Other studies on laparoscopic and robotic pyelolithotomy have observed no postoperative peritonitis or infection [24,25]. In the removal of kidneys affected by XGP, our study revealed that the transperitoneal approach did not increase the risk of postoperative peritonitis or intra-abdominal abscess formation.

Escherichia coli (51.4%), Proteus mirabilis (31.4%), Klebsiella pneumonia (5.7%), and Staphylococcus aureus (2.7%) were the most prevalent species detected in the urine and abscess bacterial cultures. Proteus mirabilis is a highly prevalent pathogen that causes renal stone infection; however, it is uncommon in patients with peritonitis [26]. Laparotomy can result in a large field of vision during surgery, direct exposure to the lesion, a large amount of normal saline irrigation, and unobstructed open drainage after surgery, which might lead to the low risk of intra-abdominal infection; this assumption requires further research to be supported.

To the best of our knowledge, this is one of the few studies comparing the two approaches to laparotomy. However, this study has several limitations. First, because of the nature of retrospective studies, our findings should be interpreted cautiously. Second, because XGP is a relatively rare disease treated through several surgical treatment modalities and because the number of patients included in each subgroup was low, the accuracy of the incidence of each outcome may have been affected. Third, this study was conducted in Taiwan’s largest medical center. The diversity of surgical methods that the surgeons perform and the generalizability of the results should be investigated in subsequent studies.

## 5. Conclusions

Traditional open surgery is a safe option for patients with XGP who require surgery. The surgical procedures lead to similar complication rates and surgical results. Patients who are overweight and expected to be suitable candidates for surgery may benefit from a midline transperitoneal approach, which does not increase the risk of postoperative peritonitis and intraabdominal infection. The flank retroperitoneal approach may also enable rapid oral intake.

## Figures and Tables

**Figure 1 jcm-11-04476-f001:**
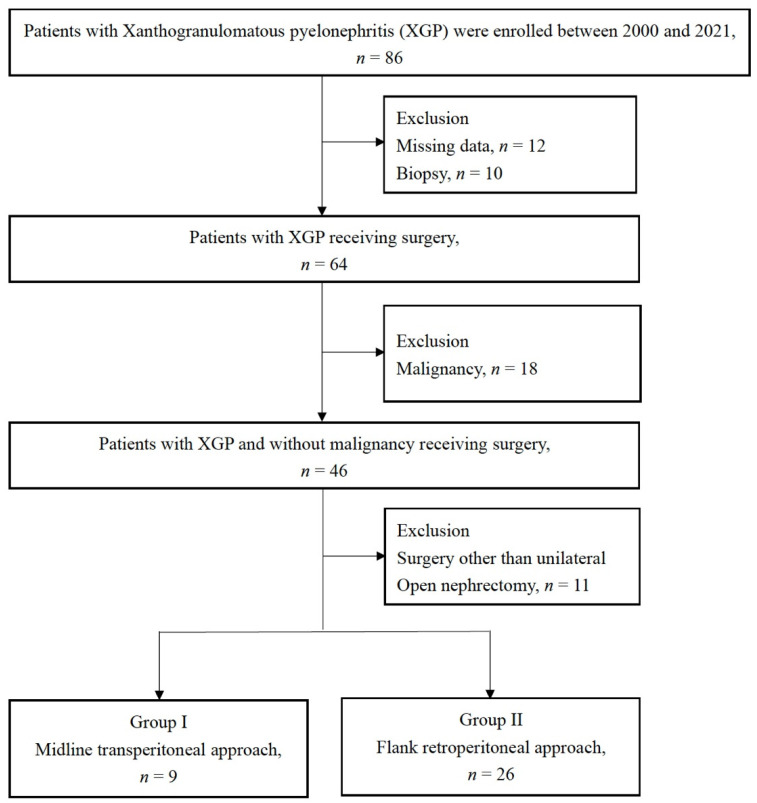
Flowchart of patient enrollment**.**

**Table 1 jcm-11-04476-t001:** Demographic and clinical characteristics of patients.

Variable	All n = 35	Group I n = 9	Group II n = 26	*p*-Value
Female, N (%)	31/35 (88.5%)	8/9 (88.9%)	23/26 (88.4%)	0.73
Age, years old (mean ± SD)	62.0 ± 9.8	61.3 ± 10.6	62.2 ± 9.5	0.43
BMI, kg/m^2^ (mean ± SD)	26.0 ± 4.3	28.3 ± 3.2	25.7 ± 4.5	0.05
Comorbidity				
Hypertension, n (%)	17/35 (48.6%)	6/9 (66.7%)	9/26 (34.6%)	0.19
Diabetes mellitus, n (%)	15/35 (42.8%)	5/9 (55.6%)	10/26 (38.4%)	0.31
Hemodialysis, n (%)	2/35 (5.7%)	1/9 (11.1%)	1/26 (3.8%)	0.45
ASA Classification (mean ± SD)	2.6 ± 0.5	2.89 ± 0.3	2.54 ± 0.6	0.02
Preoperative laboratory data				
White blood cell, /mm^3^ (mean ± SD)	9757 ± 3510	9525 ± 944	9850 ± 4106	0.41
Hemoglobin, g/dL (mean ± SD)	9.3 ± 1.4	9.5 ± 1.3	9.2 ± 1.5	0.36
Creatinine clearance, ml/min (mean ± SD)	62.1 ± 35.2	62.8 ± 47.6	61.8 ± 28.0	0.49
Initial presentations				
Urinary tract infection, n (%)	23/35 (65.7%)	6/9 (66.7%)	17/26 (65.3%)	0.64
Abdominal pain, n (%)	27/35 (77.1%)	7/9 (77.8%)	20/26 (76.9%)	0.67
Hematuria, n (%)	15/35 (42.9%)	2/9 (22.2%)	13/26 (50.0%)	0.14
Fever, n (%)	12/35 (34.3%)	3/9 (33.3%)	9/26 (34.6%)	0.64
Preoperative diagnosis				
Radiographic XGP, n (%)	4/35 (11.4%)	0/9 (0.0%)	4/26 (15.4%)	0.29
Renal calculus, n (%)	24/35 (68.6%)	3/9 (33.3%)	21/26 (80.8%)	0.02
Hydronephrosis, n (%)	23/35 (65.7%)	5/9 (55.6%)	18/26 (69.2%)	0.36
Renal mass, n (%)	8/35 (22.9%)	5/9 (55.6%)	3/26 (11.5%)	0.02
Malek and Elder classification (mean ± SD)	2.63 (0.54)	3.0 (0)	2.5 (0.57)	<0.05
Preoperative drainage placement, n (%)	14/35 (40.0%)	3/9 (33.3%)	11/26 (42.3%)	0.47

BMI, body mass index. ASA, American Society of Anesthesiologists. XGP, xanthogranulomatous pyelonephritis.

**Table 2 jcm-11-04476-t002:** Perioperative and postoperative results.

Variable	All n = 35	Group I n = 9	Group II n = 26	*p*-Value
All cause of 30-day mortality, n (%)	0/35 (0.0%)	0/9 (0.0%)	0/26 (0.0%)	N/A
Estimated blood loss (EBL), ml (mean ± SD)	737.1 ± 606.8	950.0 ± 679.9	663.5 ± 560.8	0.13
Blood transfusion during surgery, n (%)	19/33 (57.6%)	8/9 (88.9%)	11/24 (45.8%)	0.28
PRBC, U (mean ± SD)	3.2 ± 2.7	4.4 ± 3.2	2.7 ± 2.3	0.09
FFP, U (mean ± SD)	1.8 ± 2.2	2.2 ± 2.7	1.6 ± 2.0	0.28
Operative duration, minutes (mean ± SD)	250.9 ± 96.6	322.3 ± 114.3	226.2 ± 75.3	0.03
Kidney specimen volumn, grams (mean ± SD)	388.4 ± 237.7	531.9 ± 292.7	338.7 ± 191.9	0.39
Intraoperative complications, n (%)	2/35 (5.7%)	1/9 (11.1%)	1/26 (3.8%)	0.45
Postoperative complications, n (%)	5/35 (14.3%)	2/9 (22.2%)	3/26 (11.5%)	0.38
Clavien–Dindo classification greater than three, n (%)	4/35 (11.4%)	1/9 (11.1%)	3/26 (11.5%)	0.73
Duration of postoperative drainage placement time, day (mean ± SD)	10 ± 7.3	11.6 ± 9.4	9.4 ± 6.2	0.28
Duration of postoperative analgesic use, day (mean ± SD)	16.9 ± 6.6	17.9 ± 7.3	16.6 ± 6.4	0.33
Duration of postoperative antibiotics, day (mean ± SD)	16.1 ± 7.6	16.3 ± 8.4	16.0 ± 7.3	0.47
Time to first oral diet, day (mean ± SD)	3.3 ± 1.5	4.6 ± 1.6	2.8 ± 1.3	0.01
ICU admission, n (%)	4/35 (11.4%)	1/9 (11.1%)	3/26 (11.5%)	0.73
Length of hospitalization after surgery, day (mean ± SD)	11.3 ± 7.2	12.9 ± 8.9	10.7 ± 6.3	0.41
Return to ER in 3 months, n (%)	1/35 (2.9%)	0/9 (0.0%)	1/26 (3.8%)	0.74

N/A, Not Applicable. PRBC, packed red blood cells. FFP, fresh frozen plasma. SD, standard deviation. ICU, intensive care unit. ER, emergency room.

**Table 3 jcm-11-04476-t003:** Pathogens detected in preoperative urine and perinephric abscess cultures.

Pathogen	All n = 35	Group I n = 9	Group II n = 26
Negative culture results, n (%)	4 (11.4%)	1 (11.1%)	3 (11.5%)
Bacteroides fragilis, n (%)	3 (8.6%)	1 (11.1%)	2 (7.7%)
Citrobacter amalonaticus, n (%)	1 (2.9%)	0 (0.0%)	1 (3.8%)
Enterobacter aerogenes, n (%)	2 (5.7%)	1 (11.1%)	1 (3.8%)
Escherichia coli, n (%)	18 (51.4%)	7 (77.8%)	11 (42.3%)
Klebsiella pneumoniae, n (%)	2 (5.7%)	1 (11.1%)	1 (3.8%)
Proteus mirabilis, n (%)	11 (31.4%)	2 (22.2%)	9 (34.6%)
Pseudomonas aeruginosa, n (%)	2 (5.7%)	2 (22.2%)	0 (0.0%)
Streptococcus agalactiae, n (%)	6 (17.1%)	0 (0.0%)	6 (23.0%)
Staphylococcus aureus, n (%)	1 (2.9%)	0 (0.0%)	1 (3.8%)
Viridans streptococcus	2 (5.7%)	0 (0.0%)	2 (7.7%)
Candida albicans, n (%)	1 (2.9%)	0 (0.0%)	1 (3.8%)
Multiple organisms, n (%)	14 (40.0%)	3 (33.3%)	11 (42.3%)

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
