# Peer review of "Surgery for Xanthogranulomatous Pyelonephritis: A Comparison of Midline Transperitoneal and Flank Retroperitoneal Laparotomy Approaches to Nephrectomy"

_jcm, 2022, doi:10.3390/jcm11154476_

Round 1
Reviewer 1 Report
This is a good retrospective analysis of XGP cases. The research question was relevant and was well addressed by a simple and clear methodology.
Though there is a comparison between open vs minimal access surgery for this entity, the comparison between the two open approaches is less documented.
The authors have designed a clear methodology and presented the results in a clear and easy way to comprehend. The conclusion is well supported by the results. The manuscript is written in a good scientific manner.
Author Response
Response to reviewer one comments:
Thanks so much for your comments and suggestions. XGP is a clinical disease with few cases, but it is a great challenge for clinicians to encounter such instances. Furthermore, the benefits of minimally invasive surgery apply only to suitable candidates. However, some patients, particularly those with rugged or bulky tumors, are not eligible candidates; conventional laparotomy is the best option, and this is the primary purpose of this research.
Finally, we would like to express our highest thanks to you for reviewing our manuscript.
Reviewer 2 Report
An interesting report of cases of surgery approaches for xanthogranulomatous pyelonephritis if presented.
Methods section are simple and easy to understand
The results section demonstrated no differences in the two approaches except for operation time and time to first oral diet.
The main perioperative outcomes are presented.
Limitation of the study is a small sample size, but as we know this is a rare disease.
Author Response
Response to reviewer 2 comments:
I really appreciate your feedback and ideas. Although there aren't many cases of XGP, it can be very difficult for doctors to run into one. Additionally, the advantages of minimally invasive surgery only apply to qualified patients. Conventional laparotomy is the ideal option for most cases, particularly those with challenging or large tumors, and this is the main goal of our research.
Finally, we would like to express our highest thanks to you for reviewing our manuscript.
Reviewer 3 Report
well written study a comparison of midline transperitoneal and flnaks retroperitoneal laparotmy approaches to nepherectomy on surgery for xanthogranulamtous pyelonephritis. however some flaws are seen. What about the laparascopic approaches in this cases? and in the study case numbers (n:9) are not enough such a this kind of comparison of two groups.
Author Response
Response to reviewer 3 comments:
Thanks so much for your feedback and some queries raised.
Point 1: What about the laparoscopic approaches in these cases?
Response 1:
The main focus of this study is on the comparison of laparotomy for XGP disease. Although there aren't many cases of XGP, it can be difficult for clinicians to encounter such instances. Additionally, the advantages of minimally invasive surgery only apply to qualified candidates. Conventional laparotomy is the optimal option, particularly for those with complicated or large tumors, and this is the primary goal of our research. During our 21-year follow-up, only 4 cases underwent laparoscopic minimally invasive surgery. The main reason is that most of these cases are quite difficult surgical cases (mainly M&E classification three patients.) Therefore, they were not included in our study.
Pont 2: In the study case numbers (n:9) are not enough such a this kind of comparison of two groups.
Response 2:
Because XGP is a rare inflammatory disease often associated with high morbidity and mortality, the number of cases discussed in the published literature is minimal, ranging from 9 to 41 cases to the best of our knowledge. In our 21-year medical records follow-up, we identified 86 people with XGP; however, we omitted several cases that were unsuitable for discussion to maintain the study's objectivity. Finally, we gathered 35 patients in total who matched the inclusion criteria. Although we agree that the number of mid-line approaches patients is too few, this is a true reflection of our study data. However, by utilizing biostatistical testing techniques, it is still possible to identify the variations in the treatment results between the two groups.
Finally, we would like to express our highest thanks to you for reviewing our manuscript and expect our work to be suitable for publication.
Round 2
Reviewer 3 Report
none
Author Response
We appreciate you giving us such a wonderful submitting experience, and we hope our work will inspire clinicians taking care of XGP patients. Hopefully, our revised version of the manuscript will meet your journal's expectations and allow our manuscript to be considered for publication.